child development; child mental health; global mental health; mental well-being

**Author for correspondence:**
Christina A. Laurenzi,
Email: christinalaurenzi@sun.ac.za

# Critical life course interventions for children and adolescents to promote mental health

Christina A. Laurenzi[1] [ID], Sihle Mamutse[1] [ID], Marguerite Marlow[1], Tatenda Mawoyo[1], Linnea Stansert Katzen[1], Liliana Carvajal-Velez[2,3], Joanna Lai[4], Nagendra Luitel[5], Chiara Servili[6], Moitreyee Sinha[7] and Sarah Skeen[1,8]

[1]Institute for Life Course Health Research, Stellenbosch University, Cape Town, South Africa; [2]Division of Data, Analytics, Planning and Monitoring, Data and Analytics Section, UNICEF, New York, NY, USA; [3]Department of Global Public Health, Karolinska Institutet, Stockholm, Sweden; [4]Programme Group, Health Section, UNICEF, New York, NY, USA; [5]Transcultural Psychosocial Organization Nepal, Kathmandu, Nepal; [6]Department of Mental Health and Substance Use, World Health Organization, Geneva, Switzerland; [7]citiesRISE, New York, NY, USA and [8]Amsterdam Institute for Social Science Research, Faculty of Social and Behavioural Sciences, University of Amsterdam, Amsterdam, Netherlands

## Abstract

Childhood and adolescence are key developmental periods in the life course for addressing mental health, and there is ample evidence to support significant, increased investment in mental health promotion for this group. However, there are gaps in evidence to inform how best to implement mental health promotion interventions at scale. In this review, we examined psychosocial interventions implemented with children (aged 5–10 years) and adolescents (aged 10–19 years), drawing on evidence from WHO guidelines. Most psychosocial interventions promoting mental health have been implemented in school settings, with some in family and community settings, by a range of delivery personnel. Mental health promotion interventions for younger ages have prioritised key social and emotional skills development, including self-regulation and coping; for older ages, additional skills include problem-solving and interpersonal skills. Overall, fewer interventions have been implemented in low- and middle-income countries. We identify cross-cutting areas affecting child and adolescent mental health promotion: understanding the problem scope; understanding which components work; understanding how and for whom interventions work in practice; and ensuring supportive infrastructure and political will. Additional evidence, including from participatory approaches, is required to tailor mental health promotive interventions to diverse groups' needs and support healthy life course trajectories for children and adolescents everywhere.

## Impact statement

While there is excitement about increased investment potential in the mental health of children and adolescents at this global moment, there is less consensus about the most effective, scalable approaches to support positive mental health. A chief consideration is how to frame interventions to respond to critical developmental periods in childhood and adolescence, and to promote mental health at this stage of the life course as a core foundation for other health behaviours. This review examines existing psychosocial interventions to promote mental health in children aged 5–10 years and adolescents aged 10–19 years, building on the evidence that informed recent World Health Organization guidelines. We identify common approaches used across different ages and population sub-groups, and how these may respond to shifts along the life course. We then distil four cross-cutting areas affecting child and adolescent mental health promotion, including measurement approaches, identifying core components, exploring what works in practice across diverse contexts, and how to ensure political will to scale up solutions. These considerations share some practical directions for future research and policy in this critical field. Engaging children and adolescents in co-developing and tailoring psychosocial interventions is one way to maximise impact. Integrating life course approaches within mental health promotive interventions can also safeguard mental health along a longer trajectory and support healthy trajectories for children and adolescents.

## Introduction

There is a growing global interest in prioritising the promotion of mental health for children, adolescents, and young people (The Lancet Child and Adolescent Health, 2021). Childhood and adolescence are important developmental periods for addressing mental health: nearly half of all

mental disorders emerge before the age of 14, with the majority, 75%, by age 25 (Kessler et al., 2005).

Critical life events – including entering puberty, moving away from home, and experiencing early pregnancy and parenthood – may contribute to shifts in mental health and well-being (Teruya and Hser, 2010; Halfon et al., 2018). Additionally, from a socioecological perspective, many children and adolescents are experiencing multiple stressors that can increase or accelerate their vulnerability to poor mental health. Shocks stemming from the COVID-19 pandemic include disruptions to education and missed opportunities for social development (Pfefferbaum, 2021; Almeida et al., 2022), linked with experiences of ongoing economic precarity and family bereavement (Hillis et al., 2021). A recent global survey of children and youth found that the climate crisis is fuelling high levels of climate anxiety and associated complex emotions (Hickman et al., 2021). Furthermore, large numbers of youth are experiencing conflict, forced displacement and migration (Scharpf et al., 2021), alongside high levels of bullying and cyberbullying (Zhu et al., 2021). These factors impede mentally healthy transitions from childhood to adolescence and adulthood, which in turn establish essential foundations for healthy societies.

However, numerous protective influences can promote child and adolescent mental health. Social support, including through peer networks and broader social connectedness, has been found to foster resilience and support better mental health in early life (Bauer et al., 2021). Emerging research speaks to the role that hopefulness might play in promoting children's and adolescents' well-being (Kirby et al., 2021). Opportunities for physical activity, as well as for learning and developing core social and emotional skills, can also have lasting, positive effects as children grow up (World Health Organization and UNICEF, 2021). While children's life circumstances and contexts may vary, these protective influences emerge as universally important.

With growing interest in this specific field, there are emerging opportunities for increased investment and responsive programming to reach more children and adolescents (World Health Organization, 2017; World Health Organization and UNICEF, 2021). However, there is a lack of clarity on where these investments are best placed, and how they might efficiently promote mental health and reduce the global burden of disease in this age group. It is clear that mental health experiences differ by context and resource availability: most research on child and adolescent mental health is based within high-income settings, despite the fact that 90% of the world's young people live in low- and middle-income countries (Galagali and Brooks, 2020) and are often more substantially affected by global crises and shocks that can inhibit positive mental health. Shifting from a disease-based, clinical focus to more holistic, comprehensive approaches may better serve a broader group of youth. Additionally, more researchers and practitioners are designing and delivering interventions to promote mental health across the life course (Halfon et al., 2018).

Life course approaches involve prioritising key developmental stages, and aligning children's and adolescents' needs with evidence-based programming responding to these transitions. Recent work has sought to clarify what life course health interventions are, and to identify core characteristics they should incorporate (Box 1). Going beyond purely technical, siloed interventions, this emerging field aims to ensure that interventions are (1) attuned to health transitions and longer-term developmental trajectories; (2) collaboratively designed to reflect and address multiple levels of children's ecosystems and (3) focused on systems and relationships that support optimal health outcomes and equity (Russ et al., 2022).

Few existing interventions – especially those for child and adolescent mental health – adopt all of these recommended domains. Yet there is a growing need to consider how interventions can be framed to respond to critical periods in childhood and adolescence and promote mental health as a core foundation for other health behaviours (Patalay and Fitzsimons, 2018; Russ et al., 2022).

Considering this life course perspective, we aimed to review a body of emerging evidence for improving child and adolescent mental health, with a distinct focus on the earliest part of the Institute of Medicine's mental health continuum: promotion (Institute of Medicine Committee on Prevention of Mental Disorders, 1994). This narrative review explores psychosocial interventions that can support positive outcomes for children (aged 5–10 years) and adolescents (aged 10–19 years), and focuses on four priority areas for ongoing research in this area: (1) understanding the prevalence and the scope of mental health conditions; (2) understanding which components work; (3) understanding how, and for whom, interventions work in practice; and (4) ensuring supportive infrastructure and political will to intervene at scale.

## Interventions for children (5–10 years old)

There are ample interventions for children aged 5–10 years aimed at preventing anxiety and disruptive behaviours. A review of reviews conducted for the recently updated WHO mhGAP guidelines on psychosocial interventions for children in this age range found small, non-significant effects in interventions to prevent emotional problems, with the majority of universally delivered interventions are based in schools, delivered by teachers or other school staff, with some additional integration in family systems (Laurenzi et al., 2022).

The evidence for programmes that teach strategies to promote positive mental health and well-being is more limited, but promising. Mental health promotion interventions for children aged 5–10 years have focused on the development of key skills for social and emotional development, including emotion regulation and/or self-regulation, as well as prosocial skills. The capacity for self-regulation can set important foundations for later interpersonal competencies (Sanders and Mazzucchelli, 2013; Sanders et al., 2019). The ability to self-regulate can also be critical in managing negative or unwanted emotions and promoting overall health and well-being (Pandey et al., 2018; Robson et al., 2020). Studies of curriculum-based, family-based, and social skills interventions in this age group have focused on improving parenting practices and

---

**Box 1.** Key provisions of life course interventions, adapted from Russ et al. (2022)

1. aimed at optimising health trajectories,
2. developmentally focused,
3. longitudinally focused,
4. strategically timed,
5. designed to address multiple levels of the ecosystem where children are born, live, learn and grow,
6. vertically, horizontally and longitudinally integrated to produce a seamless, forward-leaning, health-optimising system,
7. support emerging health development capabilities,
8. collaboratively codesigned by transdisciplinary research teams, including stakeholders,
9. incorporate family-centred,
10. strengths-based,
11. antiracist approaches, and
12. focus on health equity.

sibling relationships and conflict resolution to improve self-regulation (Pandey et al., 2018). Physical activity and physiological stress responses – including yoga, breathing, guided mindfulness and other cognitive-behavioural approaches – have been found to support children's capacities for sustained attention, and promote cognitive and physiological awareness in children related to the development of resilience (Lynch et al., 2004; Wyman et al., 2010; Mendelson et al., 2018). For prosocial behaviours, the Good Behavior Game (GBG) intervention uses a reward-based component for children to promote appropriate on-task behaviours. A recent meta-analysis shows that the GBG had a significant positive impact on peer-rated conduct problems compared to controls in a classroom evaluation (Smith et al., 2021).

More targeted approaches may be important in specific cases. Psychological interventions that aim to develop effective coping skills and emotion regulation have been shown to improve children's mental health in the aftermath of war and in refugee contexts (Tyrer and Fazel, 2014). However, few limited trials examine the efficacy of interventions focused on prevention, promotion and treatment in this population (Morina and Sterr, 2019). Interventions with children in humanitarian settings may involve more therapeutic approaches; for example, existing cognitive-behavioural therapy interventions have a greater impact on symptoms of post-traumatic stress disorder than on depression and anxiety (Lawton and Spencer, 2021).

Furthermore, for both universal and targeted populations in this age group, leveraging parental/caregiver support can be crucial to intervention success (Hajal and Paley, 2020). Because parents and/or caregivers play pivotal roles in the lives of their children, parenting skills interventions can promote positive mental health in their children. Emotion-related parenting practices include management, discussion and coaching of children's emotional responses, all of which can enhance children's emotional vocabulary. Positive, nurturing, non-abusive parenting practices can encourage self-regulation in children (Sanders et al., 2019). A recent randomised controlled trial of Parenting for Lifelong Health for Young Children – a parental skills programme designed for parents of children ages 2–9 years old in low-resource settings – found that the programme enhanced positive parenting and reduced harsh parenting in a South African site (Ward et al., 2020).

### Interventions for adolescents (10–19 years)

Interventions for adolescents aged 10–19 years present additional opportunities for supporting foundational skills that can promote positive mental health at a pivotal life stage (World Health Organization, 2017). While substantial evidence has focused on preventing mental health disorders in this age group – both among universal populations and among adolescents with emerging conditions – more interventions have begun to focus on mental health promotion and adopting strengths-based approaches across populations (Onyeka et al., 2022).

Existing interventions for adolescents have tended to focus on developing similar skills as interventions for younger children – such as self-regulation and coping skills – while engaging additional skills that include problem-solving, goal setting, cognitive-behavioural processes and interpersonal and assertiveness skills (Skeen et al., 2019). These specific skills can support an adolescent's developmental transition to adulthood given increasing autonomy and identity formation; they can also be protective against poor mental health. Because social environments are critical to

adolescents' own sense of well-being, interventions that draw on practical examples and address broader social and contextual factors may be particularly important (van der Westhuizen et al., 2022). Inclusive, gender-responsive approaches are also increasingly important in reflecting adolescents' emerging autonomy and identities (World Health Organization and Europe., 2011). Interventions delivered to adolescents can provide opportunities to practice skills at a time when they may be encountering situations where they can apply these skills; they may also be further tailored for younger and older adolescents depending on the intended outcomes and approaches (Skeen et al., 2020).

These interventions can be effective across a range of populations and settings. A 2019 systematic evidence review, conducted for the inaugural WHO guidelines on adolescent mental health under the *Helping Adolescents Thrive* initiative, provided a critical overview of psychosocial interventions for adolescents ages 10–19 years old (Skeen et al., 2018; World Health Organization, 2020). For adolescents, universally-focused interventions tend to be school-based, engaging strategies that support positive mental health, expand opportunities for mental health literacy and protect against a wide range of mental health risk factors. These interventions often rely on teachers or school staff as implementers for the sake of cost and convenience, but can also employ lay personnel or community members (Collins et al., 2014; Skryabina et al., 2016). Parents and caregivers can also be engaged, sometimes in parallel or together with adolescents in specific sessions (Skeen et al., 2020).

Mental health interventions for special groups have also engaged promotive elements with adolescents who are likely to be more prone to poor mental health based on their risk profile – such as adolescents in humanitarian settings, those who are pregnant or parenting, and those living with HIV (Skeen et al., 2018; World Health Organization, 2020). These interventions may address adolescents' specific needs, or integrate context-tailored strategies to promote better mental health. Diverse delivery personnel have been used to deliver these interventions based on the targeted group and their stated needs; for example, peer-delivered interventions for adolescents living with HIV can reduce stigma and foster trust (Mark et al., 2019; Laurenzi et al., 2022). This field is expanding, with relatively good evidence on strategies to support positive mental health and broader areas to target.

### Cross-cutting issues for researching and implementing child and adolescent mental health interventions

In this section, we distil four priority areas that span across children and adolescent mental health intervention research and identify gaps in research and practice that remain.

### *Understanding the prevalence and the scope of mental health conditions*

Investing in accurate measurement of child and adolescent mental health is essential to understand the burden of poor mental health, to ensure early detection and treatment, and to prioritise relevant services and programmes to improve child and adolescent mental health in meaningful ways (Hayes et al., 2021). Despite the importance of mental health, measurement efforts for children and adolescents are complicated by several factors, necessitating thoughtful planning and flexible measurement approaches, including cultural adaptation of the tools used for data collection with this group.

First, the timing of measurement is important, given clear age-related patterns in mental health and differences in disease burden profiles between children, younger adolescents and older adolescents (Guthold et al., 2021). Tools that are appropriate and valid for children and younger adolescents, including those with more severe mental health challenges, are extremely limited. Recent work has highlighted the insufficiency of screening and assessment items for suicidal ideation and risk in children aged 5–11 years, who represent a small but growing proportion of suicides (Ayer et al., 2020).

Second, the focus of measurement is also important, given the wide spectrum of mental disorders, understanding and experiences of mental health by age, gender, culture and geography. Focusing on epidemiological data and psychopathology alone may miss the diversity and depth of experiences with mental health, especially for children and adolescents. Additionally, these developmental periods are often characterised by a variety of co-occurring risk factors, which require specific measurement considerations. Engaging children and adolescents with lived experience in formative work, and later development and adaptation activities, can provide essential insights on context, experience and acceptability (Reed et al., 2021).

Lastly, few valid, reliable tools to measure positive aspects of mental health during childhood and adolescence are available or feasible across a range of contexts, cultures and populations (Erskine et al., 2017). Ideally, these tools should enable both service providers and researchers to engage with children and adolescents in their home language, using local definitions and understandings of mental health symptoms. Self-report measurement approaches provide an opportunity for children and adolescents to be involved in the assessment process (De Vries et al., 2018), but may also be shaped by social desirability bias, when participants consciously or subconsciously over- or under-state an issue to conform to perceived social norms (Simkiss et al., 2021). This issue is particularly complex given the stigma associated with mental health. Self-reports might be triangulated with other sources of data; for instance, caregivers and teachers can also complete observational assessments of the child or adolescent (De Vries et al., 2018). As observational approaches are typically more costly and resource-intensive, however, participatory methods may be an alternative, promising mode of engaging children while collecting accurate data (Fenwick-Smith et al., 2018). In low-resource settings, brief self-report screening measures are frequently used to estimate the scope of the problem, or to make decisions about mental health policy and practice. This can be problematic, since using screening tools with inappropriate validity risks under or overestimating the burden of mental health problems, both of which have negative consequences for research and programming efforts (Kohrt and Kaiser, 2021).

Using appropriate, validated tools and correctly interpreting the results is therefore central to improving global child and adolescent mental health measurement.

Recent work by the Measurement of Mental Health Among Adolescents at the Population Level (MMAP) initiative, spearheaded by UNICEF and supported by WHO and other global partners, has sought to narrow this gap (Carvajal et al., 2021). MMAP has supported extensive validation work to fill this need with adolescents aged 10–19 years, and its protocol provides guidance to conduct cultural adaptation of tools to make them suitable for specific settings (Carvajal et al., 2022). Such data and evidence are urgently needed to guide strategic actions allowing for timely, effective mental health prevention and treatment efforts to improve outcomes across the life course.

## Understanding which components work

A second cross-cutting priority is to identify components of effective interventions – to distil core "building blocks" and build consensus around what works across a range of settings (Chorpita et al., 2005; Wolpert et al., 2021). These approaches can be descriptive, such as common elements analyses (Boustani et al., 2015; Boustani et al., 2020), or more quantitative in nature in examining which components predict better outcomes (Melendez-Torres et al., 2015).

As part of the WHO's adolescent mental health guidelines, our team meta-analysed components of universal interventions, seeking components that could consistently predict larger effect sizes across outcomes at different timepoints (Skeen et al., 2019). Components were adapted from the PracticeWise manual (PracticeWise, 2009). Seven consistent components emerged: interpersonal skills, emotional regulation, alcohol and drug education, mindfulness, problem-solving, assertiveness training and stress management. We found that two of these, interpersonal skills and emotional regulation, were associated with larger effect sizes for positive mental health outcomes. These components have been integrated into WHO's guidelines for adolescents as well as the *Helping Adolescents Thrive* (HAT) toolkit, supporting the implementation of these guidelines (World Health Organization and UNICEF, 2021).

Other recent work, including a body of research funded by the Wellcome Trust Active Ingredients Commission, has explored a more broadly conceptualised set of components that can promote improved mental health. On a more individualised basis, these include components such as self-compassion (Egan et al., 2022) and engagement with the arts (Easwaran et al., 2021). Effective components may also be societal or structural in nature, either as interventions themselves or in broader contexts. This type of component may include access to green spaces and recreational areas (Reece et al., 2021), or increased neighbourhood cohesion (Donnelly et al., 2016).

By advancing our understanding of which components predict better mental health outcomes, we can move to examine the most promising ways to integrate these components into psychosocial interventions. Importantly, this components work should be continually updated, and also seek to draw across diverse interventions implemented across geographical contexts, enabling a more robust understanding of these components.

## Understanding how, and for whom, interventions work in practice

Building from a common understanding of effective components, disentangling the processes underlying these interventions helps us understand the pathways and mechanisms that can lead to specified outcomes. This priority spans theory-building work, implementation science, and longitudinal approaches to evidence generation.

Realist reviews can provide frameworks for understanding underlying theories – describing how one or more interventions work in practice, for which groups, and under which circumstances (Pawson et al., 2004). The review process is inductive and iterative in nature; researchers aim to identify demi-regularities, or semi-predictable patterns of behaviour that re-occur, to examine how these might lead to outcomes (Wong et al., 2010). Contexts and mechanisms are explored as they map onto outcomes, and ultimately researchers can develop, test, and refine theories about how interventions work, leading to a more explanatory style of analysis. A recent realist review examined psychosocial interventions that

used emotion regulation skills, finding a number of provisions that supported reduced symptoms of depression and anxiety among adolescents (Skeen et al., 2020). These include tailoring interventions to target populations (age, developmental stage, gender, culture, neurodiversity, mental health status); selecting skilled, well-trained, empathic and supported practitioners to implement the intervention; and providing opportunities for skills reinforcement and parental/caregiver engagement outside of the intervention context (Pote, 2021). Gender and age differences in intervention impact may also emerge within these analyses, as well as differences within groups exposed to adversities.

These findings can be further bolstered by integrating implementation science approaches into intervention evaluation. While implementation science methods may be used alongside more traditional evaluation methods – including measures such as participant attendance, fidelity to the intervention, quality of delivery, participant feedback or cost-effectiveness – these data are rarely used to drive changes in intervention design or delivery (Bauer and Kirchner, 2020). In identifying more suitable ways to reach children and adolescents, to measure unanticipated context- or systems-related factors, and to streamline intervention delivery, researchers can maximise the impact of interventions and enable theories to be testable. Relatedly, engaging children and adolescents in the intervention design and evaluation process can be especially effective (Ozer et al., 2018). Youth preferences must be well-integrated into interventions, especially for adolescents but even for child-focused interventions. Sustained engagement can be facilitated through using co-development processes that integrate and elevate adolescent perspectives, establishing adolescent advisory boards that oversee and/or govern a research project along its full continuum, and creating protocols to ensure youth engagement is central to research project planning, decision-making and dissemination (Oliveras et al., 2018; Ozer et al., 2018).

A final key area for assessing how, and for whom, interventions work in practice is through harnessing longitudinal data. Just as more foundational research is needed to understand how mental health trajectories shift along the life course, we need to be able to measure the effects of psychosocial interventions on a longer time horizon. Early childhood interventions may show short-term protective effects, but these impacts may be overwhelmed by social and environmental risks to child development (Tomlinson et al., 2020), especially for populations experiencing multiple adversities. Longitudinal intervention cohorts can provide more granularity as to how intervention effects may be attenuated or strengthened over time – as well as insights into critical life course events at distinct ages. These analyses could pinpoint critical moments in children's and adolescents' mental health trajectories where shifts in skills may be needed and may provide important evidence for tailoring strategy by age or mental health experience. These data can also provide important evidence about if and how effectively children and adolescents are able to apply intervention skills over time.

### Ensuring supportive infrastructure and political will to intervene at scale

On a broader level, identifying ways to prioritise child and adolescent mental health promotion as part of larger systems is essential. Mental health is ultimately a cross-cutting issue; however, it remains conceptualised as a health issue in global and national development agendas.

Policies that support child and adolescent mental health are critical to promoting mental health and preventing disorders (Mokitimi et al., 2018; Skokauskas et al., 2018). While this imperative is echoed in global agendas such as the Sustainable Development Goals, globally, over 40% of countries have no policies that include mental health, and only 10% of countries have policies that include children and adolescents (World Health Organization, 2021b). This misalignment is even greater in LMICs, which often lack resources for "non-essential" areas of health or rely on foreign funding to support their national budgets (Babatunde et al., 2021). The required resources can include trained personnel, adequate facilities and appropriate assessment tools. This oversight can also happen in places where policies do exist: policies may be comprehensive but lack the infrastructure, resources or political will to be effectively implemented or integrated into practice (Mokitimi et al., 2018).

Importantly, few policies conceptualise mental health on a broad spectrum: elevating the significance of mental health promotion alongside prevention and treatment is also a critical priority for expanding access. One way of effectively shifting the framing is to engage stakeholders across sectors including education, social development and labour, in addition to health. Recent guidance, including the HAT Toolkit, has identified key actions for partners from various sectors to engage in mental health promotion collaboratively (World Health Organization and UNICEF, 2021). Approaches such as task-shifting can also be valuable, especially for delivering psychosocial interventions, in strengthening the mental health workforce to reach more individuals at low cost (McInnis and Merajver, 2011).

Supportive infrastructure for mental health can be fostered at the level of the home and school environments, where children and adolescents spend most of their time (Wang and Degol, 2016). In addition to the parental/caregiver engagement approaches shared above, which aim to influence how parents interact with their children to promote good mental health, school engagement is an additional critical aspect. The WHO's Health Promoting Schools framework offers a roadmap for integrating health promotion into educational spaces, advocating a whole-school approach (World Health Organization, 2021a). Practically, this process might involve incorporating health into the school curriculum, changing the social and physical environment of school, and involving families and communities in reinforcing key messages (Langford et al., 2017). School climate interventions may reduce depressive symptoms (Singla et al., 2021), decrease suicidality (La Salle et al., 2017) and enhance resilience (Riekie et al., 2017) – suggesting that positively altering school environments can have significant outcomes. Beyond these immediate environments, there is a need for sustained engagement at country level to unlock multi-level advocacy and funding – all of which have the capacity to advance youth mental health initiatives (Hayes et al., 2021).

### Discussion

As children, adolescents, and youth under the age of 25 account for 42% of the world's population – approximately 2.6 billion individuals – there is a clear imperative to find ways to promote mental health in a sustainable and accessible way (Clark et al., 2020; UN, 2022). This review presents important directions for continuing research and programming with this group, laying out cross-cutting priorities and persistent gaps.

Focusing on early intervention for child and adolescent mental health is critical, along with building additional evidence on mental

health promotion strategies for universal, targeted and indicated populations (Skeen et al., 2019). Emerging evidence shows that life course approaches may be best positioned to provide holistic, contextually-responsive interventions to promote health in young populations; however, there is limited work currently integrating core principles of life course research in mental health-focused interventions (Tomlinson et al., 2022) as well as for special groups. Interventions should also be responsive to how critical life events sequence over time and how transitions layer on one another. By focusing interventions on the stages and events that are most likely to create adult illness, health trajectories can change before symptoms emerge (Halfon et al., 2022; Russ et al., 2022).

Increased interest in investing in youth mental health means that more opportunities to implement programmes may soon be available. At a larger scale, specific guidance exists for country-level teams and international organisations seeking to implement well-supported programming to promote and protect adolescent mental health (World Health Organization and UNICEF, 2021). There is a need to ensure that interventions are responsive to social and structural factors as well as to unexpected shifts (Petagna et al., 2022). This responsiveness includes critically evaluating the cultural relevance of existing intervention approaches, and appraising the individual and collaborative capacities of stakeholders across multiple sectors to implement and integrate policies (Mokitimi et al., 2019). Cost-effectiveness studies or economic evaluations are also much-needed, to enable more calculated decisions about where and how to best invest financial and human resources. There is also a need to expand ongoing work within LMICs. Beyond gathering larger-scale epidemiological data, there is a need for good-quality, independent intervention research to reinforce investments in psychosocial programmes for youth in LMICs facing a slew of challenges such as lack of mental health education and awareness, low political urgency, stigma, limited data and a dearth of mental health resources (Zhou et al., 2020).

The COVID-19 pandemic has also given countries an opening to re-envision child and adolescent mental health programmes and policies. It is essential to continue to identify ways in which the pandemic has created new needs and also prompted innovative responses around child and adolescent mental health (Panchal et al., 2021). Promoting mental health in this age group as the COVID-19 pandemic evolves relies on including children and adolescents in conversations around their needs and their own well-being, and continuing to utilise emerging modalities – such as digital support, involving family members, and task-shifting to peers – to engage them.

## Conclusion

To advance developments in this field, interventions for children and adolescents that are evidence-based, tailorable, and adaptable should be prioritised. These solutions should be suitable to implement in low-resource settings, and deliverable by non-specialists. They should also reach adolescents where they are – in schools or other child- and adolescent-friendly spaces. It is critical that youth themselves are engaged in the development and dissemination of these solutions (Mawn et al., 2015). Recent evidence for life course approaches to promote child and adolescent health is promising. Building on this emerging field, and the cross-cutting priorities in this review, more work should seek to develop and test appropriate life course approaches to maximise impacts with this critical age

group. Ultimately, these approaches have greater potential to centre the rights of children and adolescents and promote healthy futures for them.

**Open peer review.** To view the open peer review materials for this article, please visit http://doi.org/10.1017/gmh.2022.58.

**Author contributions.** C.A.L., S.M., T.M., L.S.K., J.L., N.L., M.S. and S.S. contributed to the conceptualisation of this review. C.A.L. led the drafting of the manuscript, with vital contributions from S.M., T.M., L.S.K., M.M., L.C., C.S. and S.S. All authors approved the final version of this manuscript.

**Financial support.** This research received no specific grant from any funding agency, commercial or not-for-profit sectors.

**Competing interest.** The authors declare none.

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
