## [Reviewer Report]

10 August 2022

To the editors of Cambridge Prisms: Global Mental Health:

We are pleased to submit our invited review article entitled, “Critical life course interventions for children and adolescents to promote mental health” for your consideration in the launch of Cambridge Prisms: Global Mental Health.

While there is excitement about increased investment potential in the mental health of children and adolescents at this global moment, there is less consensus about the most effective, scalable approaches to support for diverse populations. A chief consideration is how to frame interventions to respond to critical periods in childhood and adolescence and promote mental health as a core foundation for other health behaviours.

In this manuscript, we review and critically evaluate the existing evidence on psychosocial interventions that promote mental health among children and adolescents, building from work we conducted in preparation for the development of two related World Health Organization guidelines. We identify psychosocial interventions for children (aged 5-10) and adolescents (10-19) delivered across a diverse range of settings, and examine core social and emotional skills that these interventions target, including for children and adolescents from different groups. We then highlight four cross-cutting areas in this field, with key gaps and considerations for expanding the field and framing interventions more clearly around life course approaches. 

This article reflects an ongoing collaboration among colleagues working across four continents and from academic, non-governmental, and policy fields—all of whom have made significant efforts to advance research and practice in child and adolescent mental health. We believe that the launch of Cambridge Prisms: Global Mental Health is an ideal platform for the issues we explore in this review and a suitable fit for the journal’s intended readership.

We have submitted this manuscript to Cambridge Prisms: Global Mental Health only, and we guarantee that no other submissions will be made elsewhere while it is under consideration. Furthermore, we have not published this work elsewhere, and should it be published in Cambridge Prisms: Global Mental Health, it will not be published elsewhere in the future, in similar form or verbatim, without permission of the editors. All authors are responsible for the reported research. The specific contributions of the authors have been detailed in the online submission portal, and all authors have approved this manuscript as submitted. 

Thank you for considering our article for publication in this exciting special issue and we look forward to receiving your comments on our work. 

Sincerely, 

Dr. Christina A. Laurenzi, on behalf of the authors

Institute for Life Course Health Research

Department of Global Health, Faculty of Medicine and Health Sciences

Stellenbosch University

Tygerberg, South Africa

---

## [Reviewer Report]

*Comments to Author*: This is a well-considered article on the broad-stroke issues related to a life course approach to interventions for children and adolescents. While the Ms speaks to these issues in large measure around a global context, comments such as "there is also a need to expand work within LMICs." (Pp24, line 471-472) give the impression that the work is largely speaking to developed contexts (unintended). It is readily acknowledged that it may be difficult to speak of developed and developing contexts in a "broad-strokes" review, but it does appear to introduce the unintended effect of foregrounding developed contexts. This is a question I often found myself grappling with in reading the excellent high-level summary of what works. For example, do common elements have equal validity in the two contexts? Is the reality of greater psychosocial and environmental risk (as suggested by the reference to children growing up in a war context or in other adverse circumstances) dissipated in using such an approach? Would the same elements be required to be embedded over longer periods of time (assuming that structural change is difficult), and if so, what would this look like?

These points of criticism should not detract from an otherwise well-written article that is structurally sound and which highlights the most significant issues which serve as benchmarks for future work. These include identifying what we know about a common elements approach in developing interventions for children and adolescents; the importance of parental/ caregiver support; self-regulation and coping skills and universal and targeted interventions. Of great utility is highlighting what research needs to focus on going forward, viz., measurement and screening, understanding which components work and building consensus around what works (taking into account my earlier comment about context), and understanding pathways and mechanisms that influence specific outcomes.

It would be especially helpful if effective pathways for change are also considered at a systems level as much of the review work focuses on individual outcomes, though the section on policy is very relevant and appropriate. A focus on implementation science as a method of intervention is especially welcomed and would tie in appropriately with looking at whole systems. It goes without saying that the engagement of youth in research is pivotal to these efforts and some examples of best practices would be helpful in this regard. Finally, examining these issues within a longitudinal framework as emphasized by the authors' is not only important from an outcomes perspective but would greatly assist in understanding critical life course events for both children and adolescents.

---

## [Reviewer Report]

*Comments to Author*: This is a very well written paper that will be extremely useful for the field as it comprehensively summarises the status quo evidence and highlights ways forward in policy, research and practice.

Here are a few comments that I think might be helpful to address:

p13 'Understanding the scope of the problem': the contents in this section refer specifically to understanding the prevalence and size of the problem rather than other sects of scope such as around interventions; if authors agree could the title be made more specific?

especially given the nature of the article not being a standard review it would be very helpful to clearly introduce the aims and structure of the article early on

there is hardly any mention of the economic evidence both in terms of cost-effectiveness of interventions as well as affordability and return-on-investment of scaling intervention - this should be highlighted as a gap

under ' cross-cutting issue for research ...' the authors include a section at the end on 'Ensuring supportive infrastructure and political will to intervene at scale' - I think this is much broader than research and seems to be more policy and practice issues and recommendations - I wonder if this needs to be renamed

---

## [Reviewer Report]

10 August 2022

To the editors of Cambridge Prisms: Global Mental Health:

We are pleased to submit our invited review article entitled, “Critical life course interventions for children and adolescents to promote mental health” for your consideration in the launch of Cambridge Prisms: Global Mental Health.

While there is excitement about increased investment potential in the mental health of children and adolescents at this global moment, there is less consensus about the most effective, scalable approaches to support for diverse populations. A chief consideration is how to frame interventions to respond to critical periods in childhood and adolescence and promote mental health as a core foundation for other health behaviours.

In this manuscript, we review and critically evaluate the existing evidence on psychosocial interventions that promote mental health among children and adolescents, building from work we conducted in preparation for the development of two related World Health Organization guidelines. We identify psychosocial interventions for children (aged 5-10) and adolescents (10-19) delivered across a diverse range of settings, and examine core social and emotional skills that these interventions target, including for children and adolescents from different groups. We then highlight four cross-cutting areas in this field, with key gaps and considerations for expanding the field and framing interventions more clearly around life course approaches. 

This article reflects an ongoing collaboration among colleagues working across four continents and from academic, non-governmental, and policy fields—all of whom have made significant efforts to advance research and practice in child and adolescent mental health. We believe that the launch of Cambridge Prisms: Global Mental Health is an ideal platform for the issues we explore in this review and a suitable fit for the journal’s intended readership.

We have submitted this manuscript to Cambridge Prisms: Global Mental Health only, and we guarantee that no other submissions will be made elsewhere while it is under consideration. Furthermore, we have not published this work elsewhere, and should it be published in Cambridge Prisms: Global Mental Health, it will not be published elsewhere in the future, in similar form or verbatim, without permission of the editors. All authors are responsible for the reported research. The specific contributions of the authors have been detailed in the online submission portal, and all authors have approved this manuscript as submitted. 

Thank you for considering our article for publication in this exciting special issue and we look forward to receiving your comments on our work. 

Sincerely, 

Dr. Christina A. Laurenzi, on behalf of the authors

Institute for Life Course Health Research

Department of Global Health, Faculty of Medicine and Health Sciences

Stellenbosch University

Tygerberg, South Africa

---

## [Reviewer Report]

*Comments to Author*: The authors have responded adequately to the review and I have no further suggestions.